# Perfluoroalkyl and Polyfluoroalkyl Substance Detection in Brewed Capsule Coffee

**DOI:** 10.3390/foods14060980

**Published:** 2025-03-13

**Authors:** Sunhye Hwang, Soyoung Kim, Minyeong Jeon, Yongsun Cho

**Affiliations:** Food Analysis Center, Korea Food Research Institute, 245 Nongsaengmyeong-ro, Iseo-myeon, Wanju-gun 55365, Jeollabuk-do, Republic of Korea; hsh0514@kfri.re.kr (S.H.); sykim@kfri.re.kr (S.K.); minyeong.jeon@kfri.re.kr (M.J.)

**Keywords:** perfluoroalkyl and polyfluoroalkyl substances, QuEChERS, liquid chromatography–tandem mass spectrometry, capsule coffee, food packaging

## Abstract

As food packaging materials are in direct contact with the food we eat and cook under heat or pressure, consumers are apprehensive of their adverse effects on the food products. Perfluoroalkyl and polyfluoroalkyl substances (PFASs) are often used in food packaging because of their hydrophobic properties; however, some PFASs are carcinogens, thus prompting further studies on their effects. In this study, a pretreatment method of 31 PFASs in coffee was established using the QuEChERS extraction method and analyzed by liquid chromatography–tandem mass spectrometry. We brewed 32 types of capsule coffee distributed in Korea, analyzed them for PFASs, and evaluated their safety. The results show that perfluorooctanoic acid and 8:2 fluorotelomer sulfonate levels are higher in machine-brewed capsule coffee than in capsule coffees brewed manually through a paper filter. However, the hazard quotient and excess cancer risk for all coffee samples are lower than the World Health Organization standards, and therefore, these samples are considered safe. The results of this study may aid in expanding the existing literature on PFAS detection in relation to human health.

## 1. Introduction

Perfluoroalkyl and polyfluoroalkyl substances (PFAS) are environmentally persistent organic pollutants, which are characterized by their typical structure of a polyfluorinated hydrophobic alkyl chain at one end and a hydrophilic functional group such as sulfonate or carboxyl at the other end [1]. The alkyl chains typically consist of 4–18 polyfluorinated carbons [2,3,4]. Perfluoroalkyl carboxylic acids (PFCAs) and perfluoroalkane sulfonic acids (PFSAs) are categorized by chain length based on the carbon number. According to the Organization for Economic Cooperation and Development, short-chain compounds pertain to PFCAs with seven or fewer carbons and to PFSAs with five or fewer carbons, whereas long-chain compounds correspond to PFCAs with eight or more carbons and PFSAs with six or more carbons [5,6]. PFAS bioaccumulation increases with increasing carbon chain length, whereby PFSAs have higher bioaccumulation than PFCAs for the same chain length [7,8].

For decades, PFASs have been used in various formulations as surfactant intermediates (e.g., firefighting foams and aircraft hydraulic fluids) and waterproofing agents in textiles (e.g., clothing and carpets) and household products (e.g., paper and nonstick cookware) [9,10]. Fluorinated paper is used as a preservative in the fast food industry in materials such as popcorn bags, bakery bags, disposable tableware, and dairy box liners [11]. PFASs are also used as processing aids in food packaging, mold-release agents, and flow agents for inks [9]. PFASs can migrate into food through contact between food and the packaging materials [12]. The migration potential of PFASs into food depends on several factors, including the type of food, contact temperature and duration, and chain length of the PFAS [12,13].

PFASs are released into the environment during manufacturing processes. These substances can accumulate in soil, drinking water, and air; thus, through the food chain, humans eventually become exposed [14,15,16]. Perfluorooctanoic acid has also been detected in surface water (0.47–19.20 ng∙L^−1^) and seawater (4.84–65.70 ng∙L^−1^) [17]. Some of the PFASs are known to be potentially harmful to humans, persist in the environment, and have the potential for bioaccumulation and biological magnification through the food chain [18,19,20]. In vivo and in vitro results indicate that PFASs exhibit toxicity that disrupts the reproductive, developmental, hepatic, neurological, immunosuppressive, and endocrine systems of laboratory animals [21,22]. Humans are exposed to PFASs through ingestion, inhalation, and hand-to-mouth transmission in contaminated areas [23,24]. Although PFASs have been linked to obesity, cancer, and immunosuppression in children [21,25], research on their toxicological effects in humans is limited. The International Agency for Research on Cancer (IARC) has classified perfluorooctanesulfonic acid (PFOS) as possibly carcinogenic to humans (Group 2B) and perfluorooctanoic acid (PFOA) as carcinogenic to humans (Group 1), highlighting the need for further investigation into their potential health risks [26].

Although PFASs are highly persistent and toxic, they are still used in food packaging because of their hydrophobic properties. Previous studies have reported PFAS migration from food packaging into food products, particularly under heat and prolonged storage conditions [27,28]. However, research specifically examining the migration of PFAS from coffee capsule packaging into brewed coffee remains limited. A coffee capsule comprises aluminum or polypropylene capsules with coffee powder and is brewed in a machine for convenience. During brewing, water at a high temperature (over 60 °C) and under pressure passes through the capsule [29]. The exposure to increased temperature prompts the query on the quantity of PFAS in brewed coffee. The migration of PFAS from the capsule into coffee is likely influenced by factors such as temperature, contact duration, and material composition. A better understanding of these parameters is necessary to assess the safety of coffee capsules comprehensively. In this study, we measured the PFAS content in brewed coffee obtained from 32 commercially available capsule coffees. As a control, we collected coffee powder from capsule coffee and brewed it through a paper filter for comparison. Unlike previous studies that primarily focused on PFAS contamination in drinking water or processed foods, this study specifically investigates the direct migration of PFAS from capsule packaging into brewed coffee under real-world brewing conditions. The results of this study provide a good indication of the safety of coffee capsules in terms of their PFAS content and contribute to expanding the existing literature on PFAS exposure through food and beverage consumption.

## 2. Materials and Methods

### 2.1. Chemicals and Reagents

In this study, 31 PFAS standards were purchased from Cambridge Isotope Laboratories (Tewksbury, MA, USA), and N-methyl-perfluorooctanesulfonamidoethanol and N-ethyl-perfluorooctanesulfonamidoethanol were purchased from Toronto Research Chemical Inc. (Toronto, ON, Canada). Isotopically labeled internal standards, as a solution of 24 chemical compounds at a concentration of 100 ng·mL^−1^ per species, were purchased from Wellington Laboratories (Guelph, ON, Canada). The PFAS standards were diluted in methanol to a concentration of 50 μg·mL^−1^ each, and the 31-species mixture was diluted further in methanol to 100 and 10 ng·mL^−1^ in 15 mL polypropylene tubes. The internal standard was diluted with methanol to 10 ng·mL^−1^ in a 15 mL polypropylene tube. All solutions were sealed and refrigerated at 4 °C until further use, and all prepared solutions were used within one month.

Ammonium acetate, formic acid (95%), anhydrous MgSO_4_, and NaCl were purchased from Sigma-Aldrich (St. Louis, MO, USA). All solvents were of liquid chromatography–mass spectrometry (LC–MS) grade and were purchased from Supelco (Bellefonte, PA, USA). The QuEChERS original extraction kit (P/N 5982-5550, Agilent Technologies, Santa Clara, CA, USA) was obtained from Agilent Technologies, and the QuEChERS method AOAC 2007.01 was used.

### 2.2. Experimental Samples

In this study, 32 capsule coffees were purchased from online suppliers. We selected 10 coffee brands from 15 different countries of origin (either single-origin or blend) in three different capsule materials (aluminum, polypropylene, and polybutylene terephthalate).

The countries of origin and packaging materials for each capsule are listed in Table 1. The capsules each contain 5–6 g of ground coffee. Capsule coffee was brewed using a coffee machine (Nespresso coffee maker/D113, China) with a water temperature of 61 °C, brewing time of 25 s, and brewing volume of 40 mL. The brewed coffee was placed in a glass, cooled to room temperature (25 °C), and transferred to a polypropylene bottle for the experiment. As a control, the ground coffee in the capsule was transferred to a funnel equipped with a paper filter and brewed with 40 mL of water heated on a hot plate to 61 °C for 25 s. The brewed coffee was placed in a glass, cooled to room temperature, transferred to a polypropylene bottle, and immediately used in the experiment.

### 2.3. Instrumentation and Experimental Conditions

Separation analysis was performed using a 1290 II LC system (Agilent Technologies, Santa Clara, CA, USA), and MS was performed using a Triple Quad 4500 system (Applied Biosystems/MDS Sciex, Framingham, MA, USA). We ensured that no background contamination occurred to increase the accuracy of the experiment (Figure 1).

The effluent from the analytical column was sent to a Triple Quad 4500 mass spectrometer equipped with a jet-stream electrospray ionization source, and the chromatographic conditions were adjusted to optimize selectivity and separation efficiency [2]. The column temperature was stabilized at 40 °C. A 5 mmol·L^−1^ aqueous solution of ammonium acetate was labeled as solvent A, and methanol was labeled as solvent B. The gradient profile is as follows: B increased from 40% to 90% in 25 min, decreased to 40% after 1 min, and was maintained for 30 min. The total LC analysis time was 30 min per sample; the solvent flow rate was fixed at 0.3 mL·min^−1^. The injection volume was fixed at 5 μL. All analytes were detected under negative ionization using the triple quadrupole multiple reaction-monitoring (MRM) mode. Nitrogen was used at 45 psi as the nebulization gas at the temperature and flow rate of 300 °C and 6 L·min^−1^, respectively, for the dry gas and 260 °C and 11 L·min^−1^, respectively, for the covering gas. The electrospray (capillary) voltage was set at 4500 V. Two MRM transitions were used to detect each compound: a higher intensity for quantification and a lower intensity for qualitative confirmation.

The detailed parameters of the MRM transitions for each standard are listed in Table 2. Under the optimal conditions, 31 standards were separated for the experiments; the chromatograms of these standards are shown in Figure 2. Table 2 provides the names, CAS numbers, and abbreviations of these 31 standards.

### 2.4. Sample Extraction and Cleanup

The coffee samples were prepared for analysis by placing 10 mL brewed coffee into a 50 mL polypropylene centrifuge tube with acetonitrile (20 mL), formic acid (150 μL), and an internal standard (200 μL; 10 μg·mL^−1^). The solution was mixed for 1 min with a vortex mixer (Barnstead International, Waltham, MA, USA). PFASs were extracted from each sample by adding 4000 mg MgSO_4_ and 1000 mg NaCl to the polypropylene tube and homogenizing it for 5 min using a multimixer (SeouLin Bioscience, Seoul, South Korea). Each sample was then centrifuged (Koki Holdings, Tokyo, Japan) for 10 min at 4500 rcf at 4 °C. The separated acetonitrile phase (10 mL) was placed into a 15 mL polypropylene centrifuge tube with MgSO₄ (1200 mg) and primary/secondary amine (400 mg) to remove polar interferences. The mixture was vortexed for 2 min to allow efficient interaction between the solvents and the sample, facilitating purification. The sample was then centrifuged at 4500 rcf for 10 min at 4 °C to separate the cleaned supernatant from residual particulates. The fractionated solution (5 mL) was filtered through a 0.22 μm nylon filter (JetBiofil, Guangzhou, China), placed in a clean centrifuge tube, and dried using an N_2_ concentrator (Cliper Life Sciences, Hopkinton, MA, USA). Although some nylon filters are known to adsorb or contain PFAS, we tested the PFAS background levels in our filters before use. The analysis confirmed that no detectable levels of PFAS were present in the blank samples filtered through the nylon filters. Therefore, we ensured that the nylon filters used in this study did not introduce any PFAS contamination. The residue was dissolved in 0.5 mL of methanol, filtered using a 0.22 μm nylon filter, transferred to a 300 μL polypropylene screw-cap micro vial (Nowlabiz, Seoul, South Korea), and analyzed by LC–tandem MS (LC–MS/MS).

### 2.5. Quality Assurance and Quantification

Before analyzing the samples, we validated the proposed analytical method through the following measurements. Linearity was assessed by preparing calibration curves using the PFAS standards (100 and 10 ng·mL^−1^) at concentrations ranging from 0.025 to 5.000 ng·g^−1^ in a coffee solution. The repeatability and extraction efficiency were measured by spiking at the three concentrations of 0.25, 0.50, and 2.50 ng·g^−1^. Matrix-matched standards were also analyzed to confirm the absence of a matrix effect. The method demonstrated high selectivity, as blank samples and spiked samples showed no significant interferences from matrix components or structurally similar compounds. The resolution between analytes was assessed and confirmed to be greater than 1.5, ensuring proper separation.

The method’s repeatability was evaluated using spiked coffee samples at different concentration levels, with relative standard deviation (RSD) values below 10% for most tested concentrations. Some compounds at low concentrations exhibited RSD values exceeding 10%, likely due to matrix effects and instrumental sensitivity variations. However, these deviations were mitigated by using stable isotope-labeled internal standards, which improved analytical precision.

The limits of detection (LOD) were calculated using the standard deviation of the PFASs measured during the recovery process and Student’s *t*-test. Seven independent analyses were performed using coffee spiked with PFAS at 0.1 ng·g^−1^. To determine a one-sided confidence interval, t-distribution was utilized based on the mean and standard deviation of replicate measurements, with the result being multiplied by the standard deviation. For the seven samples with six degrees of freedom, the t-value for a 99% confidence interval is found to be 3.14. The LOD was calculated by multiplying the standard deviation of the seven samples by 3.14, and the limits of quantification (LOQs) were calculated by multiplying the standard deviation by 10 [24].

To further verify the reliability of the method at LOQ levels, ion ratios and retention times were evaluated. The ion ratio deviations at LOQ levels were within ±20% compared to high-concentration samples (e.g., 2.5 ng/g), and the retention time difference was within 0.02 min for all target analytes. Measurement uncertainty was not evaluated in this study; however, future research should consider incorporating uncertainty estimation to further validate the analytical method.

### 2.6. Exposure and Risk Analysis

The PFAS content in coffee was determined to evaluate the risk of PFAS exposure from capsule packaging. The uncertainty in the average concentrations was minimized using a variable distribution value estimated through a Monte Carlo simulation (Crystal Ball ver. 11.1.3, 2022). Each value was replicated 10,000 times using a normal distribution for simulation purposes. The LOQs were applied at concentrations below the LOD for coffee.

The lifetime average daily dose (*LADD*) is calculated as follows [30]:(1)$$LADD\left(\frac{\frac{mg}{kg}bw}{day}\right)=\frac{C\left(\frac{mg}{g}\right)\times CR\left(\frac{g}{day}\right)}{BW\left(kg\right)},$$
where *C* is the average PFAS concentration in coffee during the exposure period, *CR* is the daily coffee intake (5.76 g·day^−1^ for adults [31]), and *BW* is the average body weight (60 kg for Korean adults [32]). The daily reference dose (*R_f_D*) for non-carcinogenic substances represents the daily dose of exposure over a lifetime that is considered safe, with no adverse effects anticipated. The hazard quotient (*HQ*) is calculated as the quotient of the human exposure value and *R_f_D* value as follows [30]:(2)$$HQ=\frac{LADD(mg/kgbw/day)}{RfD(mg/kgbw/day)},$$
where the *R_f_D* value of PFOA is 0.00002 [33]. Furthermore, the excess cancer risk (*ECR*) associated with these carcinogens is expressed as the product of the carcinogenicity (*Q*1*, oral slope factor) and *R_f_D* [34]:(3)ECR=LADD(mg/kgbw/day)×Q1(mg/kgbw/day),
where the Q1 value of PFOA is 0.07 [33].

### 2.7. Statistical Analysis

All statistical analyses were conducted using Microsoft Excel (Microsoft Corp., Redmond, WA, USA), IBM SPSS software (v.20; SPSS Inc., Armonk, NY, USA), and Monte Carlo simulation (Crystal Ball ver. 11.1.3, 2022). Descriptive statistics (mean, standard deviation, and relative standard deviation) were calculated using Excel. The normality of the data distribution was assessed using the Shapiro–Wilk test in SPSS.

For comparisons between manually brewed coffee and machine-brewed coffee, a paired *t*-test was performed if the data followed a normal distribution. If normality was not satisfied, the Wilcoxon signed-rank test was used as a nonparametric alternative. Differences in PFAS concentrations among different capsule materials were evaluated using one-way ANOVA, followed by Tukey’s post hoc test for multiple comparisons. A significance level of *p* < 0.05 was considered statistically significant for all analyses.

To estimate the uncertainty and variability of PFAS exposure levels, Monte Carlo simulation was conducted with 10,000 iterations using a normal distribution model. The results were expressed as probability distributions to assess potential exposure risks.

## 3. Results and Discussion

### 3.1. Background Contamination Trapping

PFAS was detected during pretreatment and in the LC–MS/MS instrument, even though no PFAS was injected during the experiment. The PFAS background contamination was reduced as much as possible to ensure the accuracy of the experiment.

First, LC–MS/MS was performed with pure methanol. Even though the PFASs were not intentionally introduced to the system, the LC–MS/MS results revealed considerable amounts of PFAS (circled peaks in Figure 1A). The contamination sources were identified by analyzing the PFAS content in the filter syringes, vials, and components of the instrument. The background contamination was captured to ameliorate the contamination throughout the extraction and analysis. Second, to reduce PFAS contamination in the LC system, methanol was used to rinse the mobile-phase lines. Polypropylene vials—which have a minimal impact on PFAS contamination—were used to replace the storage bottles and sample vials for LC injections. Additionally, syringes equipped with nylon filters but without Teflon rubber packing were used for sample extraction, and the nylon filters were pre-tested to confirm the absence of PFAS contamination before use. Third, to prevent contaminants from entering the sample loop, a short C18 LC analytical column (precolumn: Atlantis T3, 50 mm × 2.1 mm I.D., 5 μm particle size, Waters, Milford, MA, USA) was installed between the mixer and sample loop. Figure 1A shows the total ion chromatograms with several contaminants showing up as background interference, recorded before making the background contamination-trapping changes. PFASs were not detected after this method was applied, as observed in the total ion chromatogram in Figure 1B.

### 3.2. Method Validation

The PFAS content in coffee was analyzed using the pretreatment described in Section 2.4. Validation of the analytical method was conducted by assessing selectivity, linearity, repeatability, recovery, LOD, and LOQ to ensure the accuracy and reliability of the method. The results are presented in Table 3.

The selectivity of the method was evaluated by analyzing blank samples and spiked samples to check for matrix interferences or signal suppression/enhancement effects. No significant interferences from matrix components or structurally similar compounds were observed, confirming that the method provides adequate specificity for PFAS analysis. The resolution between analytes was assessed and confirmed to be greater than 1.5, ensuring effective separation of target compounds. Additionally, stable isotope-labeled internal standards were used to correct for matrix effects, improving quantification accuracy.

Linearity was confirmed across a concentration range of 0.025–5.000 ng·g^−1^ using an eight-point calibration curve, with correlation coefficients (r^2^) ranging from 0.9918 to 0.9999.

To evaluate method precision, repeatability was assessed by spiking coffee samples at three concentration levels (0.25, 0.50, and 2.50 ng·g^−1^). The relative standard deviation (RSD) values for repeatability were below 10% for most tested concentrations, confirming the method’s precision. Specifically, RSD values ranged from 0.7% to 30.0% at low concentrations (0.25 ng·g^−1^), 0.8% to 9.9% at medium concentrations (0.50 ng·g^−1^), and 0.6% to 10.5% at high concentrations (2.50 ng·g^−1^). While most RSD values remained within acceptable limits, some compounds at low concentrations exhibited RSD values exceeding 10%. These deviations were primarily due to matrix effects and lower instrument sensitivity at trace levels. However, applying stable isotope-labeled internal standards mitigated these variations, improving the overall precision of the method.

The accuracy of the method was further evaluated through recovery analysis, which ranged from 71.6% to 126.0%, with most compounds falling within the acceptable range of 80–120%. Some variation in recovery for specific PFASs was observed, likely due to matrix effects; however, these effects were mitigated using stable isotope-labeled internal standards.

The limits of detection (LOD) and limits of quantification (LOQ) were determined to be 0.004–0.082 ng·g^−1^ and 0.012–0.260 ng·g^−1^, respectively. To further confirm the reliability of the method at LOQ levels, ion ratios and retention times of detected PFASs were evaluated. The ion ratio deviations at LOQ levels were within ±20% compared to high-concentration samples (e.g., 2.5 ng/g), and the retention time difference was within 0.02 min for all target analytes. For example, in the case of PFOA, the quantification ion (369.0 *m*/*z*) and confirmation ion (168.9 *m*/*z*) ratio was 35.2% at the LOQ level and 34.8% at 2.5 ng/g, resulting in only a 1.1% variation. Similarly, for PFOS, the retention time at the LOQ level was 17.42 min, while at 2.5 ng/g, it was 17.44 min, showing a difference of only 0.02 min. These findings confirm that the method maintains its reliability even at LOQ levels.

To verify the absence of contamination and carryover effects, reagent blank samples were analyzed at 25-sample intervals throughout the study. The acceptance criterion was set at a signal-to-noise ratio (S/N) < 3, and no detectable levels of PFASs were observed in any reagent blank samples, confirming that the analytical system was free from background contamination. This ensures that all detected PFAS concentrations in coffee samples originated from the actual sample content rather than external contamination.

Although measurement uncertainty was not explicitly evaluated in this study, incorporating uncertainty estimation in future research would help further validate the method’s accuracy and precision.

### 3.3. Sample Analysis

ND indicates the result was not detectable (under LOD). The effect of the coffee capsule container on the detection of PFAS was experimentally determined; the results are shown in Table 4 and Figure 3. Table 4 reports the concentrations of PFOA and 8:2 fluorotelomer sulfonate (8:2 FTS) for each sample as the average of three trials, each for the machine-brewed coffee and manually brewed coffee. The percent increase in the concentration of 8:2 FTS for the machine-brewed coffee relative to the manually brewed coffee is also given.

To verify that the detected PFAS in brewed coffee did not originate from the water used in the extraction process, we conducted blank tests on the water samples 10 times. No PFAS were detected in any of the blank water samples, confirming that the detected PFAS in brewed coffee resulted from the coffee capsules rather than contamination from the water source.

In Figure 3, only PFAS concentrations above the LOQ are visualized, while values between the LOD and LOQ are recorded in Table 4 but excluded from the heatmap. PFOA was detected at concentrations between the LOD and LOQ in five manually brewed and five machine-brewed samples. Since these values do not meet the quantification limit, they were not included in the heatmap but are still available in Table 4 for reference.

In addition, 8:2 FTS was detected in all samples, with higher amounts in the machine-brewed samples (0.263–1.514 ng·g^−1^) than in the manually brewed samples (0.127–0.248 ng·g^−1^). The highest 8:2 FTS concentration was noted in sample #22 for the manually brewed samples and in sample #28 for the machine-brewed samples.

Brewing the capsule coffee in a coffee machine results in a minimum increase of 38.7% and a maximum increase of 875.3% in FTS compared to that for manual brewing, with an average increase of 245.6%. The FTS content in aluminum capsule coffee increases by 39.2–719.9% when machine brewed compared to that in manual brewing. Eight out of the twenty-three samples (35%) have a higher increase than the average of 245.6%. The PFOA content of polypropylene capsule coffee increases by 38.7–623.1% when machine brewed compared to manual brewing. Three out of six samples have higher-than-average content, with 50% of the machine-brewed coffees showing higher-than-average increases of 8:2 FTS. Polybutylene terephthalate capsule coffee increases by 141.4–875.3% when machine brewed compared to that for manual brewing, with one in three having higher-than-average increases.

Although the small number of coffee capsule samples makes it difficult to draw definitive conclusions, polypropylene capsule coffee was determined to have higher PFAS content, which is likely to be associated with the dissolution of PFAS in water at a high temperature and pressure. Manually brewed coffee tends to have a lower PFAS content than machine-brewed coffee.

Figure 3 presents a heatmap comparison of machine-brewed and manually brewed coffee using only PFAS concentrations above the LOQ. This ensures that only quantifiable data is visualized, providing a more accurate representation of PFAS migration in capsule coffee.

### 3.4. Risk Assessment

The LADD and noncarcinogenic HQ values of PFOA are 5.76 × 10^−6^ and 2.88 × 10^−1^, respectively. The World Health Organization (WHO) considers noncarcinogens safe if their HQ is less than 1 [33]. As the HQ value of PFOA is less than 1, the PFAS content in the coffee capsule is considered safe. In addition, the ECR of PFOA is 4.03 × 10^−7^, which is within the limit prescribed by the WHO (1.0 × 10^−5^), indicating that PFOA is at a safe level [33].

## 4. Conclusions

In this study, we analyzed the PFAS content in coffee extracted from commercially available capsule coffees to determine if the contaminants were present at a safe level. The samples were pretreated using the QuEChERS method, and an analytical method was developed for the simultaneous analysis of 31 PFAS standards in coffee using LC–MS/MS. The developed method was validated in terms of linearity, recovery, precision, LOD, and LOQ. Of the 31 PFASs, only PFOA and 8:2 FTS were detected in the capsule coffee samples. The PFAS content in the machine-brewed coffee is higher than that in the manually brewed coffee. Among all materials, high levels of PFASs were detected in polypropylene coffee capsules, indicating possible PFAS migration from the packaging to the food when heat and pressure were applied to the polypropylene material. Nonetheless, the risk assessment of the commercial capsule coffee shows results that meet international standards. Further research should be conducted on the migration of PFASs from food packaging and food through the analysis of PFASs in packaging materials.

## Figures and Tables

**Figure 1 foods-14-00980-f001:**
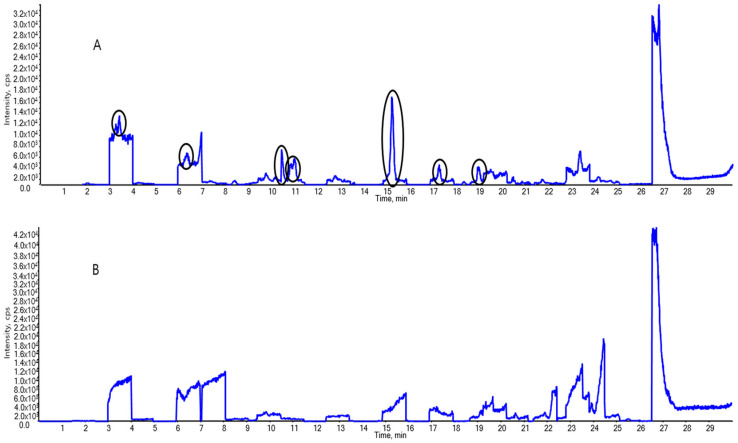
Total ion chromatograms of the background interference (**A**) before and (**B**) after application of the background contamination trapping technique. The black circled peaks are per- and polyfluoroalkyl substances (PFASs), such as PFOS and 11Cl-PF3OUdS.

**Figure 2 foods-14-00980-f002:**
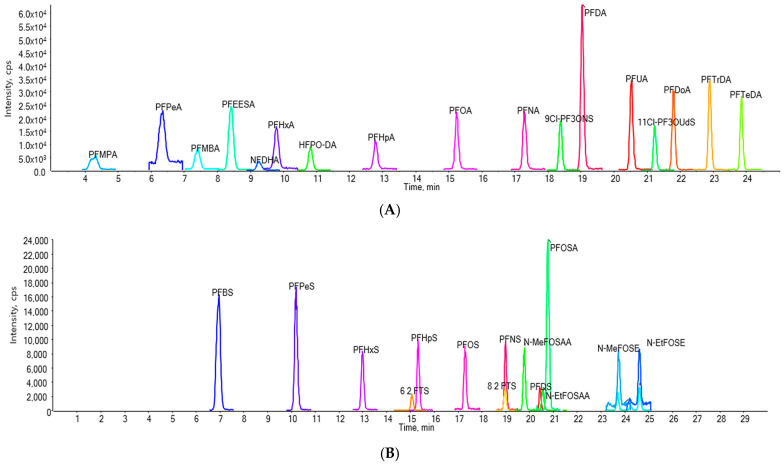
Extracted ion chromatograms of standard PFASs. (**A**) Perfluorinated carboxylic acids (PFCAs), fluoropolymers, and other PFASs. (**B**) Perfluorosulfonic acids (PFSAs), fluorotelomer sulfonates (FTSs), perfluorooctanesulfonamidoacetic acids (FOSAAs), and perfluorooctanesulfonamidoacetic acids (FOSAs). The chromatograms represent a 10 ng/mL mixed standard solution in methanol, which was spiked into the blank water used in the coffee experiment, resulting in a final concentration of 0.5 ng·g^−1^. Acronyms for all compounds are listed in Table 2.

**Figure 3 foods-14-00980-f003:**
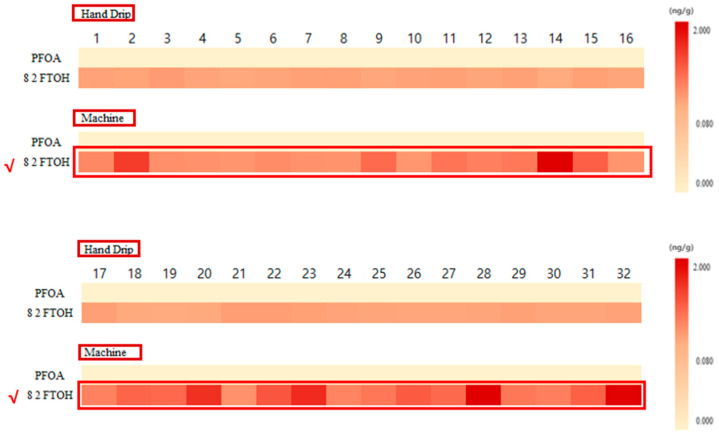
Heat map showing average PFOA and 8:2 FTOH concentrations in manually brewed and machine-brewed coffee. The heatmap includes only PFAS concentrations above the LOQ, while values between LOD and LOQ are reported in Table 4 but excluded from visualization. Heat-map comparison of 32 capsule coffees using manual brewing and machine brewing.

**Table 1 foods-14-00980-t001:** Capsule coffee origins and capsule materials.

Sample Number	Coffee Origins	Materials	
		Capsule	Lid
1	Columbia	Aluminum	Aluminum
2	Columbia	Aluminum	Aluminum
3	Columbia	Aluminum	Aluminum
4	Columbia	Aluminum	Aluminum
5	Columbia	Aluminum	Aluminum
6	Columbia	Aluminum	Aluminum
7	Columbia	Aluminum	Aluminum
8	Columbia	Aluminum	Aluminum
9	Columbia	Aluminum	Aluminum
10	Columbia	Aluminum	Aluminum
11	Columbia, Brazil, Ethiopia	Aluminum	Aluminum
12	Ethiopia	Aluminum	Aluminum
13	Ethiopia, Guatemala	Polybutylene terephthalate	Aluminum
14	Brazil, Guatemala	Polybutylene terephthalate	Aluminum
15	Ethiopia, Indonesia	Polybutylene terephthalate	Aluminum
16	Columbia, Papua New Guinea, Uganda	Aluminum	Aluminum
17	Uganda, Vietnam, Indonesia	Polypropylene	Aluminum
18	Brazil, Uganda, Vietnam	Polypropylene	Aluminum
19	Brazil, Columbia, Uganda	Polypropylene	Aluminum
20	Brazil, Columbia, Honduras	Polypropylene	Aluminum
21	Angola	Polypropylene	Aluminum
22	Brazil, Columbia, Papua New Guinea	Polypropylene	Aluminum
23	Ethiopia	Aluminum	Aluminum
24	Brazil, Ethiopia	Aluminum	Aluminum
25	Ethiopia	Aluminum	Aluminum
26	India, Costa Rica, El Salvador	Aluminum	Aluminum
27	Costa Rica, Guatemala, Ethiopia, India	Aluminum	Aluminum
28	Ethiopia	Aluminum	Aluminum
29	India, Columbia, Honduras	Aluminum	Aluminum
30	India, Ethiopia, Honduras	Aluminum	Aluminum
31	India, Columbia, Honduras	Aluminum	Aluminum
32	Costa Rica, Ethiopia, Nicaragua	Aluminum	Aluminum

**Table 2 foods-14-00980-t002:** List of per- and polyfluoroalkyl substances (PFASs) analytes and the corresponding multiple reaction-monitoring parameters for mass spectrometry detection.

Group	Compound	Abbreviation	CAS No.	Retention Time (min)	Q1 (*m*/*z*)	Q3 (*m*/*z*)	DP ^a^ (V)	EP ^b^ (V)	CE ^c^ (V)	CXP ^d^ (V)	Internal Standard	
											Compound	Abbreviation
PFCAs	Perfluoropentanoic acid	PFPeA 1	2706-89-0	6.5	262.9	219.0	–10	–10	–12	–15	Perfluoro-n-[^13^C_5_]pentanoic acid	M5PFPeA
		PFPeA 2			262.9	69.0	–10	–10	–54	–3		
	Perfluorohexanoic acid	PFHxA 1	2923-26-4	9.9	312.9	268.8	–10	–10	–14	–13	Perfluoro-n-[1,2,3,4,6-^13^C_5_]hexanoic acid	M5PFHxA
		PFHxA 2			312.9	119.0	–10	–10	–30	–5		
	Perfluoroheptanoic acid	PFHpA 1	375-85-9	12.9	362.9	319.0	–5	–10	–16	–7	Perfluoro-n-[1,2,3,4-^13^C_4_]heptanoic acid	M4PFHpA
		PFHpA 2			362.9	168.9	–5	–10	–24	–9		
	Perfluorooctanoic acid	PFOA 1	335-67-1	15.4	412.9	369.0	–5	–10	–16	–7	Perfluoro-n-[^13^C_8_]octanoic acid	M8PFOA
		PFOA 2			412.9	168.9	–5	–10	–26	–11		
	Perfluorononanoic acid	PFNA 1	375-95-1	17.4	462.9	419.0	–10	–10	–16	–7	Perfluoro-n-[^13^C_9_]nonanoic acid	M9PFNA
		PFNA 2			462.9	218.8	–10	–10	–4	–13		
	Perfluorodecanoic acid	PFDA 1	3830-45-3	19.1	512.9	469.0	–10	–10	–18	–7	Perfluoro-n-[1,2,3,4,5,6-13C_6_]decanoic acid	M6PFDA
		PFDA 2			512.9	218.9	–10	–10	–26	–11		
	Perfluoroundecanoic acid	PFUA 1	60871-96-7	20.7	562.9	518.9	–5	–10	–18	–9	Perfluoro-n-[1,2,3,4,5,6,7-^13^C_7_]undecanoic acid	M7PFUdA
		PFUA 2			562.9	268.8	–5	–10	–26	–15		
	Perfluorododecanoic acid	PFDoA 1	307-67-5	21.9	612.9	569.0	–10	–10	–18	–11	Perfluoro-n-[1,2-^13^C_2_]dodecanoic acid	MPFDoA
		PFDoA 2			612.9	319.0	–10	–10	–28	–7		
	Perfluorotridecanoic acid	PFTrDA 1	72629-94-8	23.0	662.9	619.0	–5	–10	–20	–11		
		PFTrDA 2			662.9	168.8	–5	–10	–36	–11		
	Perfluorotetradecanoic acid	PFTeDA 1	376-06-7	24.0	712.8	669.0	–5	–10	–24	–15	Perfluoro-n-[1,2-^13^C_2_]tetradecanoic acid	M2PFTeDA
		PFTeDA 2			712.8	218.8	–5	–10	–34	–9		
PFSAs	Perfluorobutanesulfonic acid	PFBS 1	29420-49-3	7.1	298.8	79.9	–55	–10	–60	–7	Sodium perfluoro-1-[2,3,4-^13^C_3_]butanesulfonate	M3PFBS
		PFBS 2			298.8	98.9	–55	–10	–56	–7		
	Perfluoropentanesulfonic acid	PFPeS 1	630402-22-1	10.3	348.9	80.0	–5	–10	–72	–5	Perfluoro-n-[1,2,3,4-^13^C_4_]heptanoic acid	M4PFHpA
		PFPeS 2			348.9	98.9	–5	–10	–64	–7		
	Perfluorohexanesulfonic acid	PFHxS 1	3871-99-6	13.1	398.9	79.9	–55	–10	–94	–7	Sodium perfluoro-1-[1,2,3-^13^C_3_]hexanesulfonate	M3PFHxS
		PFHxS 2			398.9	99.0	–55	–10	–74	–5		
	Perfluoroheptanesulfonic acid	PFHpS 1	21934-50-9	15.5	448.9	80.0	–25	–10	–106	–7	Perfluoro-n-[^13^C_8_]octanoic acid	M8PFOA
		PFHpS 2			448.9	99.0	–25	–10	–82	–7		
	Perfluorooctanesulfonic acid	PFOS 1	4021-47-0	17.4	498.9	79.9	–15	–10	–110	–5	Sodium perfluoro-1-[^13^C_8_]octanesulfonate	M8PFOS
		PFOS 2			498.9	99.0	–15	–10	–94	–7		
	Perfluorononanesulfonate	PFNS 1	98789-57-2	19.1	548.9	79.9	–25	–10	–118	–7	Perfluoro-n-[^13^C_9_]nonanoic acid	M9PFNA
		PFNS 2			548.9	99.0	–25	–10	–92	–7		
	Perfluorodecanesulfonate	PFDS 1	2806-15-7	20.6	598.8	80.0	–135	–10	–130	–5	Sodium perfluoro-1-[^13^C_8_]octanesulfonate	M8PFOS
		PFDS 2			598.8	98.9	–135	–10	–96	–7		
	6:2 Fluorotelomer sulfonate	H4-PFOS (6:2 FTS) 1	27619-94-9	15.2	427.0	407.0	–85	–10	–34	–7	Sodium 1H,1H,2H,2H-perfluoro-1-[1,2-^13^C_2_]octanesulfonate	M2-6 2FTS
		H4-PFOS (6:2 FTS) 2			427.0	80.9	–85	–10	–72	–5		
	8:2 Fluorotelomer sulfonate	H4-PFDeS (8:2 FTS) 1	27619-96-1	19.1	526.9	506.9	–95	–10	–40	–9	Sodium 1H,1H,2H,2H-perfluoro-1-[1,2-^13^C_2_]decanesulfonate	M2-8 2FTS
		H4-PFDeS (8:2 FTS) 2			526.9	81.0	–95	–10	–86	–7		
FOSAAs	N-Methyl perfluorooctanesulfonamidoacetic acid	N-MeFOSAA 1	2355-31-9	19.9	569.9	419.0	–60	–10	–30	–7	N-methyl-d_3_-perfluoro-1-octanesulfonamidoacetic acid	d3-N-MeFOSAA
		N-MeFOSAA 2			569.9	482.9	–60	–10	–24	–9		
	N-Ethyl perfluorooctanesulfonamidoacetic acid	N-EtFOSAA 1	2991-50-6	20.7	583.9	419.0	–65	–10	–28	–7	N-ethyl-d_5_-perfluoro-1-octanesulfonamidoacetic acid	d5-N-EtFOSAA
		N-EtFOSAA 2			583.9	526.0	–65	–10	–30	–11		
FOSAs	Perfluorooctanesulfonamide	PFOSA 1	754-91-6	20.8	497.9	78.0	–95	–10	–80	–7	Perfluoro-1-[13C_8_]octanesulfonamide	M8FOSA
		PFOSA 2			497.9	64.0	–95	–10	–140	–5		
	N-Methyl-perfluorooctane sulfonamidoethanol	N-MeFOSE 1	24448-09-7	23.7	615.9	59.0	–15	–10	–68	–5	Perfluoro-1-[13C_8_]octanesulfonamide	M8FOSA
	N-Ethyl-perfluorooctane sulfonamidoethanol	N-EtFOSE 1	1691-99-2	24.6	629.9	59.0	–55	–10	–56	–5		
Misc.	2,3,3,3-Tetrafluoro-2-(1,1,2,2,3,3,3-heptafluoro propoxy)propanoic acid	HFPO-DA 1	13252-13-6	10.9	328.9	168.9	–15	–10	–18	–5	Perfluoro-n-[^13^C_8_]octanoic acid	M8PFOA
		HFPO-DA 2			328.9	184.8	–15	–10	–34	–9		
	Potassium 9-chlorohexadecafluoro-3-oxanonane-1-sulfonate	9Cl-PF3ONS 1	73606-19-6	18.5	530.8	350.9	–60	–10	–38	–7	Sodium perfluoro-1-[^13^C_8_]octanesulfonate	M8PFOS
		9Cl-PF3ONS 2			530.8	35.0	–60	–10	–76	–3		
	11-Chloroeicosafluoro-3-oxaundecane-1-sulfonic acid	11Cl-PF3OUdS 1	83329-89-9	21.3	630.8	450.9	–110	–10	–44	–11		
		11Cl-PF3OUdS 2			630.8	35.1	–110	–10	–76	–9		
	Perfluoro(2-ethoxyethane)sulfonic acid	PFEESA 1	113507-82-7	8.5	314.8	134.9	–45	–10	–30	–11	Sodium perfluoro-1-[2,3,4-^13^C_3_]butanesulfonate	M3PFBS
		PFEESA 2			314.8	198.8	–45	–10	–22	–15		
	Perfluoro-4-methoxybutanoic acid	PFMBA 1	863090-89-5	7.6	279.0	84.9	–5	–10	–28	–5	Perfluoro-n-[^13^C_5_]pentanoic acid	M5PFPeA
	Perfluoro-3-methoxypropanoic acid	PFMPA 1	377-73-1	4.4	228.9	84.9	–5	–10	–28	–5	Perfluoro-n-[^13^C_4_]butanoic acid	MPFBA
	Perfluoro-3,6-dioxaheptanoic acid	NFDHA 1	151772-58-6	9.4	294.8	200.8	–5	–10	–14	–11	2,3,3,3-Tetrafluoro-2-(1,1,2,2,3,3,3-heptafluoropropoxy)-^13^C_3_-propanoic acid	M3HFPO-DA
		NFDHA 2			294.8	135.0	–5	–10	–26	–9		

^a^ Declustering potential; ^b^ Entrance potential; ^c^ Collision energy; ^d^ Collision cell exit potential.

**Table 3 foods-14-00980-t003:** Linearity, recovery, precision, LOD ^a^, and LOQ ^b^ for the extraction of PFAS from spiked coffee ^c^.

Group	Acronym	Linear Range (ng·g^−1^)	r ^d^	0.25 ng·g^−1^		0.5 ng·g^−1^		2.5 ng·g^−1^		LOD (ng·g^−1^)	LOQ (ng·g^−1^)
				RSD ^e^ (%)	Recovery (%)	RSD (%)	Recovery (%)	RSD (%)	Recovery (%)		
PFCAs	PFPeA	0.025–5	0.9994	8.1	104.3	5.7	77.4	2.5	93.7	0.068	0.218
	PFHxA	0.025–5	0.9995	2.0	97.4	3.7	96.0	4.2	95.6	0.008	0.025
	PFHpA	0.025–5	0.9986	1.5	90.1	0.8	97.9	3.7	91.7	0.014	0.045
	PFOA	0.025–5	0.9991	0.7	105.7	2.8	125.9	3.0	97.6	0.082	0.260
	PFNA	0.025–5	0.9998	4.7	78.6	3.6	80.5	2.3	93.8	0.015	0.049
	PFDA	0.025–5	0.9997	3.0	95.5	4.7	83.9	2.6	92.3	0.034	0.107
	PFUA	0.025–5	0.9996	7.6	92.1	2.3	99.3	2.8	102.0	0.027	0.086
	PFDoA	0.025–5	0.9989	2.5	92.1	3.8	89.9	0.6	92.7	0.009	0.028
	PFTrDA	0.025–5	0.9994	10.2	104.6	4.8	96.4	2.3	100.8	0.020	0.064
	PFTeDA	0.025–5	0.9992	4.8	107.7	2.5	97.6	2.7	97.6	0.008	0.024
PFSAs	PFBS	0.025–5	0.9995	0.7	91.3	0.9	89.1	3.3	96.3	0.021	0.067
	PFPeS	0.025–5	0.9989	5.2	119.0	9.9	121.8	2.1	109.7	0.020	0.063
	PFHxS	0.025–5	0.9984	4.4	93.5	3.5	89.4	1.5	89.5	0.010	0.032
	PFHpS	0.025–5	0.9991	4.5	115.8	9.3	123.0	4.1	119.9	0.005	0.015
	PFOS	0.025–5	0.9991	4.7	89.1	9.8	98.5	2.9	104.4	0.016	0.052
	PFNS	0.025–5	0.9996	3.1	110.6	6.6	99.3	1.6	99.1	0.004	0.012
	PFDS	0.1–5	0.9991	4.2	100.6	1.6	105.5	3.5	97.4	0.006	0.019
FTSs	6:2 FTS	0.1–5	0.9918	4.4	103.8	1.3	99.0	3.3	92.6	0.021	0.068
	8:2 FTS	0.1–5	0.9938	8.5	94.9	5.4	84.1	1.7	87.6	0.044	0.141
FOSAs	N-MeFOSAA	0.1–5	0.9998	7.0	105.1	8.8	104.2	3.6	126.0	0.035	0.113
	N-EtFOSAA	0.1–5	0.9981	10.8	89.7	0.8	87.4	6.9	90.6	0.024	0.076
	PFOSA	0.025-5	0.9993	8.8	90.7	6.7	85.4	3.8	95.3	0.022	0.072
	N-MeFOSE	0.05-5	0.9998	19.6	86.6	6.5	71.6	3.7	78.5	0.019	0.061
	N-EtFOSE	0.05-5	0.9992	30.0	106.1	5.9	79.8	10.5	82.9	0.019	0.060
Misc.	HFPO-DA	0.025-5	0.9987	6.1	82.3	8.2	75.7	1.3	76.9	0.006	0.019
	9Cl-PF3ONS	0.025-5	0.9991	4.3	100.1	5.4	98.5	1.3	100.9	0.016	0.051
	11Cl-PF3OUdS	0.025-5	0.9999	7.9	98.9	1.9	91.6	2.3	97.5	0.017	0.055
	PFEESA	0.025-5	0.9997	1.3	91.6	6.4	86.9	2.8	94.8	0.025	0.081
	PFMBA	0.025-5	0.9993	2.3	86.0	5.4	73.3	1.3	82.8	0.009	0.029
	PFMPA	0.025-5	0.9994	0.8	94.2	2.9	91.5	1.1	93.8	0.025	0.079
	NFDHA	0.1-5	0.9995	4.5	121.6	3.4	124.1	3.7	125.4	0.004	0.013

^a^ Limit of detection; ^b^ Limit of quantification; ^c^ n = 3 for each level; ^d^ Coefficient of correlation; ^e^ Relative standard deviation. This table presents validation parameters (linearity, recovery, precision, LOD, and LOQ). Since these values are not based on group comparisons, statistical significance (superscripts) is not applied.

**Table 4 foods-14-00980-t004:** Average PFAS concentrations according to the brewing conditions of capsule coffee (unit: ng·g^−1^).

Group	Acronym	Condition	1	2	3	4	5	6	7	8	9	10	11	12	13	14	15	16
PFCAs	PFOA	HandDrip	ND	ND	ND	ND	ND	ND	ND	ND	ND	ND	<0.26	<0.26	ND	ND	ND	ND
		Machine	ND	ND	<0.26 ^a^	<0.26 ^a^	ND	ND	ND	ND	ND	ND	<0.26 ^a^	ND	ND	ND	ND	ND
		± %	-	-	-	-	-	-	-	-	-	-	43.284	-	-	-	-	-
FTSs	8:2 FTS	HandDrip	0.180 ± 0.02 ^a^	0.178 ± 0.03 ^a^	0.227 ± 0.04 ^a^	0.175 ± 0.01 ^a^	0.150 ± 0.05 ^a^	0.164 ± 0.02 ^a^	0.187 ± 0.03 ^a^	0.193 ± 0.04 ^a^	0.156 ± 0.01 ^a^	0.179 ± 0.02 ^a^	0.189 ± 0.03 ^a^	0.159 ± 0.04 ^a^	0.190 ± 0.05 ^a^	0.127 ± 0.02 ^b^	0.195 ± 0.03 ^a^	0.160 ± 0.04 ^a^
		Machine	0.352 ± 0.04 ^a^	0.830 ± 0.05 ^a^	0.316 ± 0.03 ^a^	0.291 ± 0.02	0.265 ± 0.03 ^a^	0.331 ± 0.04 ^a^	0.285 ± 0.05 ^a^	0.275 ± 0.02 ^a^	0.544 ± 0.03 ^a^	0.263 ± 0.04 ^a^	0.487 ± 0.05 ^a^	0.410 ± 0.02 ^a^	0.458 ± 0.03 ^a^	1.236 ± 0.04 ^b^	0.612 ± 0.05 ^a^	0.268 ± 0.02 ^a^
		± %	95.591	366.892	39.207	66.027	76.970	101.674	52.012	42.747	249.684	47.084	157.126	157.116	141.351	875.253	214.210	67.294
**Group**	**Acronym**	**Condition**	**17**	**18**	**19**	**20**	**21**	**22**	**23**	**24**	**25**	**26**	**27**	**28**	**29**	**30**	**31**	**32**
PFCAs	PFOA	HandDrip	<0.26	ND	ND	ND	ND	ND	<0.26 ^a^	ND	<0.26 ^a^	ND	ND	ND	ND	ND	ND	ND
		Machine	<0.26	ND	ND	ND	ND	ND	ND	ND	ND	ND	ND	ND	ND	ND	<0.26 ^a^	ND
		± %	0.356	-	-	-	-	-	-	-	-	-	-	-	-	-	-	-
FTSs	8:2 FTS	HandDrip	0.232 ± 0.03 ^a^	0.147 ± 0.04 ^a^	0.144 ± 0.02 ^a^	0.153 ± 0.03 ^a^	0.247 ± 0.04 ^a^	0.248 ± 0.02 ^a^	0.224 ± 0.03 ^a^	0.191 ± 0.04 ^a^	0.180 ± 0.01 ^a^	0.175 ± 0.03 ^a^	0.175 ± 0.04 ^a^	0.185 ± 0.02 ^a^	0.212 ± 0.05 ^b^	0.179 ± 0.02 ^a^	0.179 ± 0.03 ^a^	0.210 ± 0.04 ^a^
		Machine	0.475 ± 0.04 ^a^	0.706 ± 0.03 ^a^	0.670 ± 0.02 ^a^	1.106 ± 0.05 ^b^	0.342 ± 0.03 ^a^	0.817 ± 0.04 ^a^	1.142 ± 0.02 ^b^	0.459 ± 0.05 ^a^	0.566 ± 0.02 ^a^	0.773 ± 0.03 ^a^	0.666 ± 0.04 ^a^	1.514 ± 0.05 ^b^	0.557 ± 0.02 ^a^	0.500 ± 0.03 ^a^	0.724 ± 0.04 ^a^	1.493 ± 0.05 ^b^
		± %	104.958	380.489	365.490	623.088	38.728	229.401	409.925	140.948	214.687	342.814	280.991	719.882	163.062	179.022	305.620	610.615

*n* = 3 for each sample; ^a^: A significant difference between HandDrip and machine methods (*p* < 0.05). ^b^: A statistically significant difference between HandDrip and machine methods (*p* < 0.01). ND: not detected. <0.26, LOQ.

## Data Availability

The original contributions presented in this study are included in the article. Further inquiries can be directed to the corresponding author.

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
