# Peer review of "Perfluoroalkyl and Polyfluoroalkyl Substance Detection in Brewed Capsule Coffee"

_foods, 2025, doi:10.3390/foods14060980_

Round 1
Reviewer 1 Report
Comments and Suggestions for Authors
The work is interesting and covers a current area which is still being studied and developed, however, in my opinion, there are some aspects that should be futher examined and detailed, expecially regarding method validation, methods description and presentation of the results.
Below I report my comments in order along the text:
line 58: it should be mentioned that the International Agency for Research on Cancer (IARC) has classified PFOS as possibly carcinogenic to humans (Group 2B) and PFOA as carcinogenic to humans (Group 1).
Line 88-92: solution stability should be reported (also in terms of time)
Line 97: it seems that a part of the sentence is missing (“from…”)
Line 119-120: the background contamination aspect should be better described
Figure 2: it is not clear if the chromatogram shown is a matrix-matched standard or not. If not I recommend to insert one.
Line 163: what does the word “purifed” means?
Line 165 and 167: some nylon filters are known to absorb PFAS or contain PFAS from the manufacturing process. Have you testes them? Moreover in line 217-218 you wrote that nylon filters weren’t used.
Line 182-185: I didn’t find the way you used to calculate LOD and LOQ in the reference reported as reference number 23.
In general these are my comments about method validation:
validation study should be detailed more and quality control measures should be added.
Parameters such as selectivity, intra-lab reproducibility and measurement uncertainty should be inserted.
No reference to the reagent blank analysis was reported in terms of frequency of analysis, acceptability criteria.
Details about the analytical batch predisposition should be inserted.
LOQ levels obtained by calculations should be tested and verified also in terms of ion ratios and retention times. The data referred to the analysis described in line 251-254 should be shown.
In general the presentation of the results should be organized in a clearer and more rational way. (for example the LOQ calculation was explained in Material and methods (line182), while linearity in Results and discussion). In my view this is quite dispersive.
Figure 3: it would be interesting to show the heat map starting from the LOQ levels.
Line 274: it is not clear if the water used to prepare the coffee (hand drip and machine) was tested.
Reviewer 2 Report
Comments and Suggestions for Authors
Review on manuscript: foods-3489205
Perfluoroalkyl and polyfluoroalkyl substance detection in brewed capsule coffee
by Sun Hye Hwang, So Young Kim, Minyeong Jeon, & Yong Sun Cho*
submitted to Foods
Research paper
This manuscript analyzed the perfluoroalkyl and polyfluoroalkyl substances profile in brewed capsule coffee. Overall, it is of great significance and well-written. Hence, I just provide some comments and suggestions about the structure and editing of this manuscript, which have been shown as follows.
Detailed recommandations:
-Materials and Methods: the "Statistical analysis" part is missing. Add it.
-What's the significance level of data difference? (p<?) How did the authors perform the analysis of variance? The information should be presented in the "Statistical analysis" part.
-Table 4: the data should be presented as a form of means ± standard deviation.
-Tables 3-4: the significance of data difference among different groups should be marked, e.g., as superscripts of a, b, c, ...
-Figure 3: looks the same with Table 4. Only one from Table 4 and Figure 3 can be kept in the manuscript, since they showed the same data.
-References: the format of all references seems disordered and ununiform. Please check and revise them one by one.
-Also References: Normaly 30-50 citied publications were approcciate for a typical research article. Suggest the authors to add more related and mostly updated references.
Round 2
Reviewer 1 Report
Comments and Suggestions for Authors
Although I have appreciated the integrations made by the authors, I still have some comments expecially regarding method validation and samples analyses.
I think that it would be useful for you the ‘Guidance document on Analytical Parameters for the determination of Per- and Polyfluoroalkyl Substances (PFAS) in Food and Feed” by EURL POPs.
In the following lines I have added a comment to some of my first observations:
Figure 2: it is not clear if the chromatogram shown is a matrix-matched standard or not. If not I recommend to insert one.
I recommend to insert also a chromatogram with matrix
Line 165 and 167: some nylon filters are known to absorb PFAS or contain PFAS from the manufacturing process. Have you testes them? Moreover in line 217-218 you wrote that nylon filters weren’t used.
Are you also sure that they do not absorb PFAS?
Line 182-185: I didn’t find the way you used to calculate LOD and LOQ in the reference reported as reference number 23.
It is not clear the reference you used for the validation design and calculations
Details about the analytical batch predisposition should be inserted.
I can’t find a proper desciption of what should be done for the routine sample analyses (batch predisposition, QC and their acceptability criteria…
LOQ levels obtained by calculations should be tested and verified also in terms of ion ratios and retention times. The data referred to the analysis described in line 251-254 should be shown.
In my opinion it’s not clear the fortification you performed to check LOQ levels experimentally.
Comments on the Quality of English Languageno comment on English for me, I'm not English native speaking
Author Response
The questions in the second review were previously raised during the first review, and we have already provided responses. Based on the line numbers, it appears that the reviewer has referred to the pre-revision version of the manuscript. Please refer to our first response for detailed answers to these points. Additionally, we have compiled the second review comments into an attached file for your reference.
